# Kilohertz waveforms optimized to produce closed-state Na⁺ channel inactivation eliminate onset response in nerve conduction block

Guosheng Yi [1,2], Warren M. Grill[1,3,4,5]*

**1** Department of Biomedical Engineering, Pratt School of Engineering, Duke University, Durham, North Carolina, United States of America, **2** School of Electrical and Information Engineering, Tianjin University, Tianjin, People's Republic of China, **3** Department of Electrical and Computer Engineering, Pratt School of Engineering, Duke University, Durham, North Carolina, United States of America, **4** Department of Neurobiology, School of Medicine, Duke University, Durham, North Carolina, United States of America, **5** Department of Neurosurgery, School of Medicine, Duke University, Durham, North Carolina, United States of America

* warren.grill@duke.edu

**Data Availability Statement:** All code and data are available via this DOI: https://doi.org/10.7924/r4z31t79k.

## Abstract

The delivery of kilohertz frequency alternating current (KHFAC) generates rapid, controlled, and reversible conduction block in motor, sensory, and autonomic nerves, but causes transient activation of action potentials at the onset of the blocking current. We implemented a novel engineering optimization approach to design blocking waveforms that eliminated the onset response by moving voltage-gated Na⁺ channels (VGSCs) to closed-state inactivation (CSI) without first opening. We used computational models and particle swarm optimization (PSO) to design a charge-balanced 10 kHz biphasic current waveform that produced conduction block without onset firing in peripheral axons at specific locations and with specific diameters. The results indicate that it is possible to achieve onset-free KHFAC nerve block by causing CSI of VGSCs. Our novel approach for designing blocking waveforms and the resulting waveform may have utility in clinical applications of conduction block of peripheral nerve hyperactivity, for example in pain and spasticity.

## Author summary

Many neurological disorders, including pain and spasticity, are characterized by undesirable increases in sensory, motor, or autonomic nerve activity. Local application of kilohertz frequency alternating currents (KHFAC) can effectively and completely block the conduction of undesired hyperactivity through peripheral nerves and could be a therapeutic approach for alleviating disease symptoms. However, KHFAC nerve block produces an undesirable initial burst of action potentials prior to achieving block. This onset firing may result in muscle contraction and pain and is a significant impediment to potential clinical applications of KHFAC nerve block. We present a novel engineering optimization

**Funding:** This work was funded by grants from the National Institutes of Health (R37 NS040894) and the National Natural Science Foundation of China (61601320). The funders had no role in study design, data collection and analysis, decision to publish, or preparation of the manuscript.

**Competing interests:** The authors have declared that no competing interests exist.

approach for designing a blocking waveform that completely eliminated the onset firing in peripheral axons by moving voltage-gated Na$^+$ channels to closed-state inactivation. Our results suggest that the resulting KHFAC waveform can generate electric nerve block without an onset response. Our approach for optimizing blocking waveforms represents a novel engineering design methodology with myriad potential applications and has relevance for the conduction block of peripheral nerve hyperactivity.

## Introduction

Multiple neurological disorders are characterized by undesirable increases in sensory, motor, or autonomic nerve activity, for example pain and spasticity. Blocking the conduction of activity in peripheral nerves has the potential to alleviate disease symptoms [1–4]. Kilohertz frequency alternating current (KHFAC) stimulation is an effective method for producing a rapid, controlled, and locally-acting conduction block [4–8]. KHFAC nerve block is quickly reversible, and thus enables both temporal and spatial control of the target tissue with minimal side effects. However, an important drawback of KHFAC block is the transient burst of action potentials (APs) activated at the onset of blocking currents [3, 6, 8–12]. This onset firing is likely to lead to intense muscle contraction and or painful sensations and is a significant impediment to potential clinical applications of KHFAC nerve block. Several approaches are proposed to reduce or eliminate the onset response, including transitioning KHFAC waveform from a high amplitude and high frequency to a low amplitude and low frequency [13], slowly increasing the amplitude of the KHFAC waveform [14, 15], introducing direct current (DC) nerve block [12, 16], and modifying electrode geometries [17]. However, these approaches are either unable to eliminate completely the onset firing or are challenging for clinical implementation [12–17]. We employed engineering optimization to design a stimulus waveform to achieve conduction block while preventing the onset responses.

Nerve excitation is regulated by voltage-gated Na$^+$ channels (VGSCs). During an AP, the VGSCs first activate from non-conducting closed states to open states when membrane depolarization exceeds a threshold. The open channels allow influx of Na$^+$ ions, which further depolarizes the cell and drives the upstroke of the AP. Membrane depolarization also causes inactivation of VGSCs, which prevents Na$^+$ influx and facilitates repolarization to the resting potential. Na$^+$ inactivation is coupled to activation [18–20], and typically occurs from the open state at strongly depolarized membrane potentials [21, 22], also called open-state inactivation (OSI). However, recordings in cardiac cells [20, 23, 24], skeletal muscle [25], neuroblastoma cells [19, 26], squid giant axon [21], medullary raphe neurons [27], and dorsal root ganglion neurons [28, 29] confirm that VGSCs can also inactivate from pre-open closed states without first opening, called closed-state inactivation (CSI). This path occurs at hyperpolarized and modestly depolarized potentials without generation of an AP [18, 22, 24, 26]. Based on these findings, we hypothesized that moving VGSCs into CSI without first causing activation would eliminate the initiation of APs and thus prevent the onset response.

The objective of this study was to design a current waveform to produce KHFAC nerve block without onset firing. We used Markov-type kinetic models [30] to simulate Na$^+$ channel CSI, and the models were previously validated with electrophysiological data from all VGSC isoforms. We first modified the McIntyre-Richardson-Grill (MRG) model [31] of a mammalian peripheral myelinated axon to incorporate Markov-type Na$_v$ 1.1 and Na$_v$ 1.6 channels. We then used particle swarm optimization (PSO) [32] first to design the profile of membrane voltage required to move the two types of VGSCs into CSI, and, subsequently, to determine the

stimulation waveform shape required to drive that voltage profile. PSO is an effective search method for dealing with non-linear optimization problems by moving the particles to a "preferred" position in their constrained domains [32]. It is effective for determining sequences of real numbers in non-linear systems [33, 34], and was well suited to designing our voltage and current waveforms. The outcome of our engineering optimization was a KHFAC current waveform that produced nerve conduction block without generating an onset response. Our results revealed that moving VGSCs into the CSI without first opening completely suppressed the onset firing and suggest that using this novel approach for designing blocking waveforms may have relevance for the conduction block of peripheral nerve hyperactivity.

## Results

### Design of transmembrane voltage trajectory to generate closed-state inactivation of VGSCs

Our first step was to design a voltage trajectory that generated CSI of VGSCs. We used a 21-node 10 μm diameter MRG myelinated axon model of mammalian peripheral nerve fiber (Fig 1A). The fast $Na^+$ ($I_{Naf}$) and persistent $Na^+$ ($I_{Nap}$) currents in each node were originally modeled using Hodgkin-Huxley (HH)-type channels [31], which represented a VGSC as an assembly of several independent gating variables and were unable to represent the dependence of $Na^+$ inactivation on activation. Since KHFAC waveforms generate a locally-acting conduction block in peripheral nerves, the original $I_{Naf}$ and $I_{Nap}$ were replaced respectively by Markov-type $Na_v$ 1.1 and $Na_v$ 1.6 channels [30] in the middle nine nodes (6 to 14) to create a hybrid model that enabled transitions of VGSCs between states. The CSI path of $Na_v$ 1.1 and $Na_v$ 1.6 channels was C1 → I1 → I2, and the OSI path was C1 → C2 → O1 → I1 → I2 (Fig 1B). The original conductance density of $I_{Naf}$ was too small for $I_{Nav11}$ to propagate faithfully an AP from node 0 to node 20, and we increased the conductance density of $I_{Nav11}$ to enable high fidelity propagation along the hybrid axon model at rates of up to 400 spikes per second (S1 Fig).

We first examined the responses in $Na_v$ 1.1 and $Na_v$ 1.6 channels when a standard voltage step (from –80 mV to –10 mV) was applied to the central node as a voltage clamp (Fig 2A). At the resting potential of –80 mV, the two VGSCs were in closed states C1. After the onset of the voltage step to –10 mV, both $Na_v$ 1.1 and $Na_v$ 1.6 channels first entered open states O1 and then inactivated (Fig 2B), i.e., OSI occurred. The magnitude of $I_{Nav11}$ and $I_{Nav16}$ reached –81.85 mA/cm$^2$ and –0.10 mA/cm$^2$, respectively. The conductance density of $Na_v$ 1.6 channels (i.e., 0.01 mS/cm$^2$) was much smaller than that of $Na_v$ 1.1 channels (i.e., 11.9 mS/cm$^2$), and $I_{Nav11}$ was much larger than $I_{Nav16}$. An AP was initiated and propagated in the hybrid model axon at the onset of voltage step (Fig 2C).

We subsequently applied PSO to design an optimized trajectory of transmembrane voltage to drive transitions along the CSI path. PSO is a search algorithm inspired by the social behavior of swarms. It finds the optimal or near-optimal solutions to numerical and qualitative problems by adjusting the trajectories of individual particles in the defined search space of an objective function [32], and is particularly applicable to solving our waveform optimization. We defined a population of 50 particles as the membrane voltages that were used as the voltage profile to control a series of voltage clamps at node 10. The objective of the PSO was to maximize the fractions of $Na_v$ 1.1 and $Na_v$ 1.6 channels in the inactivated states while minimizing the fractions in the open states. The PSO-generated voltage profile $V_{PSO}$ increased nonlinearly from the resting potential (–80 mV) to a suprathreshold level (–10 mV) over a duration of 46 ms (Fig 2D, top) and drove transitions of $Na_v$ 1.1 and $Na_v$ 1.6 channels to CSI.

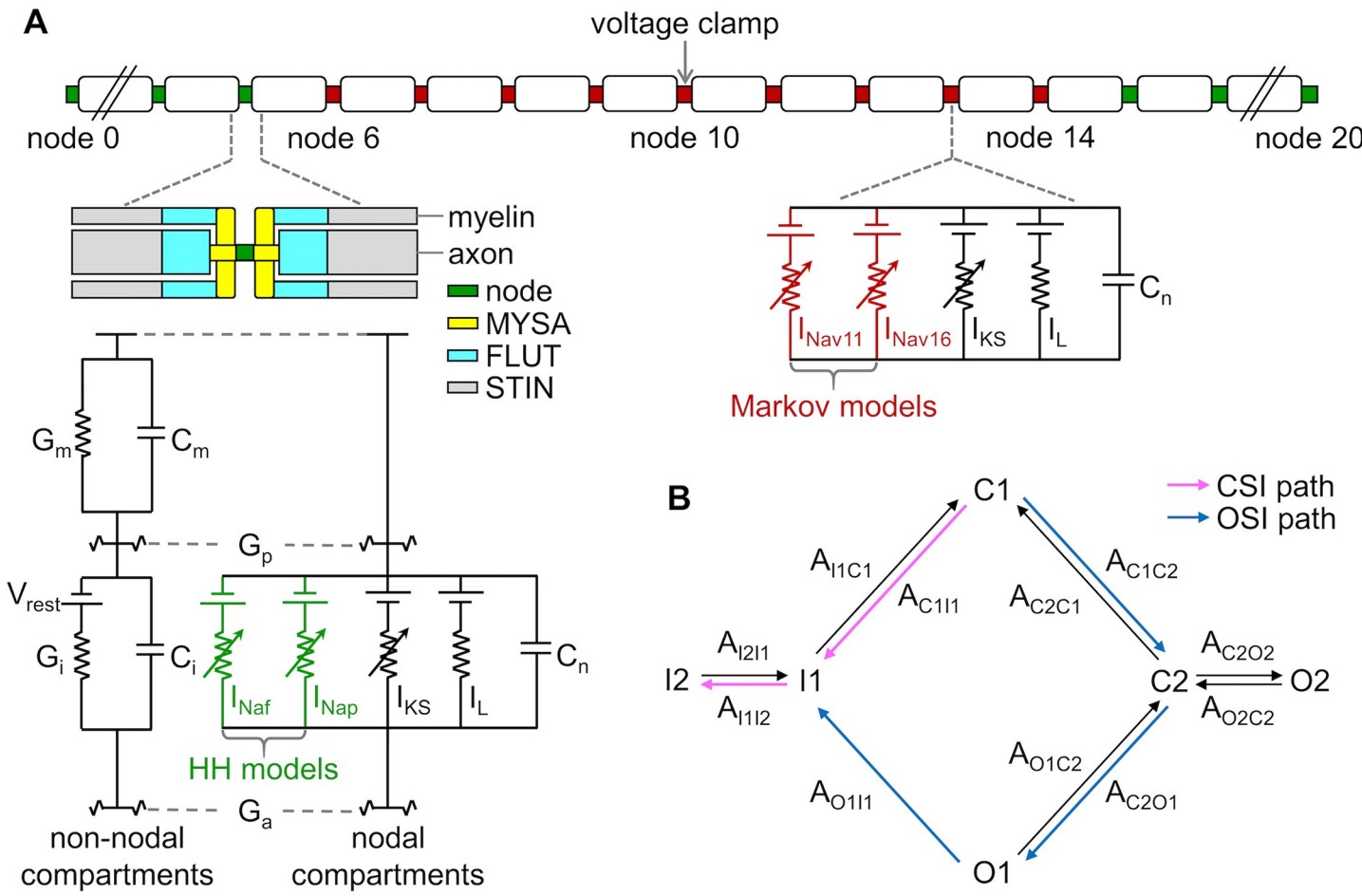

**Fig 1. Hybrid model of a peripheral myelinated axon incorporating Markov-type VGSCs. (A)** Schematic of the model, which included 21 nodes of Ranvier and 20 internodal segments. In the original MRG model, each node included fast Na$^+$ ($I_{Naf}$), persistent Na$^+$ ($I_{Nap}$), slow K$^+$ ($I_{KS}$), and linear leakage ($I_L$) currents, and membrane capacitance ($C_n$), and the ionic currents were described using HH-type kinetics. We substituted $I_{Naf}$ and $I_{Nap}$ in node 6 to node 14 with Markov-type Na$_v$ 1.1 ($I_{Nav11}$) and Na$_v$ 1.6 ($I_{Nav16}$) channels, respectively. The internodal segments used a double-cable concentric structure to represent the myelin attachment segment (MYSA), paranode main segment (FLUT), and internode segment (STIN). The internodal segments were modeled with linear conductances ($G_m$, $G_i$) and membrane capacitances ($C_m$, $C_i$). Adjacent segments were connected by axoplasmic ($G_a$) and periaxonal conductances ($G_p$). Resting membrane potential $V_{rest}$ was –80 mV, and the fiber diameter was 10 μm. The axon-specific electrical and geometrical parameters followed McIntyre *et al* [31] and are provided in S1 and S2 Tables. **(B)** Schematic of the states and transitions of Markov-type VGSCs [30]. C1 and C2 were closed states, O1 and O2 were open states, I1 and I2 were inactivation states, and $A_{w1w2}$ was the transition rate from state $w1$ to state $w2$. Pink lines represented a CSI path, and blue lines represented an OSI path. The parameters of the transition rate for Na$_v$ 1.1 and Na$_v$ 1.6 channels are provided in S3 Table.

The transitions from the closed states to the inactivated states caused by voltage profile $V_{PSO}$ were similar for Na$_v$ 1.1 and Na$_v$ 1.6 channels. At the resting potential, VGSCs were in C1 state. After depolarization, the fraction of either channel in C1 was gradually reduced (Fig 2E), the fractions in I2 increased, and there was little increase in the fraction in O1. Thus, more Na$_v$ 1.1 and Na$_v$ 1.6 channels were driven to inactivation directly from the resting state, which reduced the availability of each channel to conduct and participate in AP generation, and when the membrane voltage reached suprathreshold levels, the VGSCs were unable to enter the open states. The Na$^+$ conductance was determined by the fraction of VGSCs in the open states, and since few Na$_v$ 1.1 channels entered the O1 state during $V_{PSO}$, $I_{Nav11}$ reached only –0.62 mA/cm$^2$ (Fig 2D, bottom). In contrast to the voltage clamp step, the PSO-generated $V_{PSO}$ successfully moved both Na$_v$ 1.1 and Na$_v$ 1.6 channels to CSI, and $V_{PSO}$ delivered as a series of voltage clamps to the central node did not evoke an AP (Fig 2F).

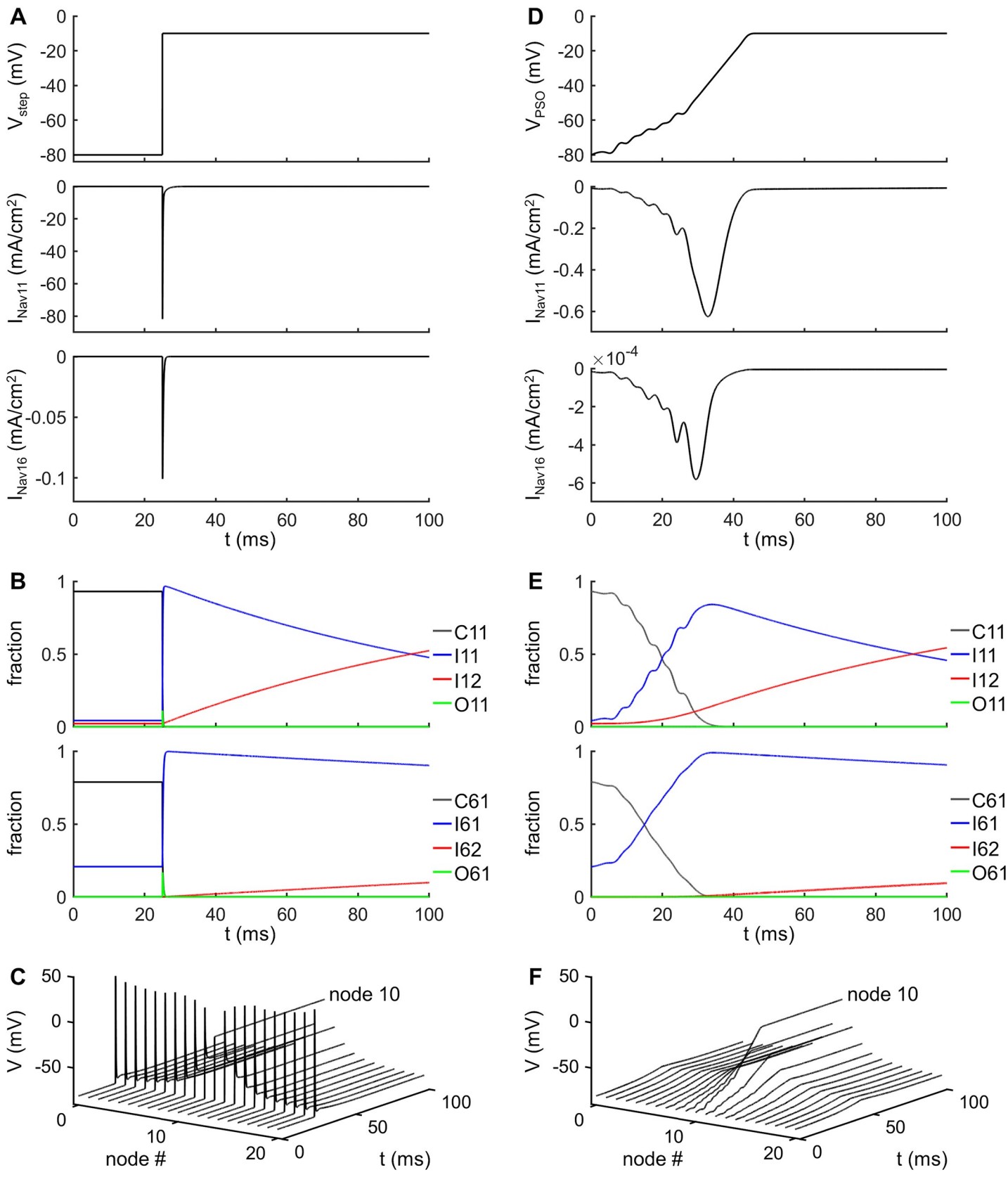

**Fig 2. Trajectory of transmembrane voltage to cause Na$^+$ CSI. (A)** Top panel: Voltage step profile $V_{step}$. The transmembrane voltage was increased from the resting potential of -80 mV to -10 mV at 25 ms. Center panel: Na$_v$ 1.1 current during $V_{step}$. Bottom: Na$_v$ 1.6 current during $V_{step}$. **(B)** Top panel: fractions of Na$_v$ 1.1 channels in states C1 (C11), I1 (I11), I2 (I12), and O1 (O11) during $V_{step}$. Bottom panel: fractions of Na$_v$ 1.6 channels in states C1 (C61), I1 (I61), I2 (I62), and O1 (O61) during $V_{step}$. **(C)** Voltage response recorded in each node when applying $V_{step}$ at node 10. **(D)** Top panel: PSO-generated optimized voltage profile $V_{PSO}$. Center panel: Na$_v$ 1.1 current during $V_{PSO}$. Bottom panel: Na$_v$ 1.6 current during $V_{PSO}$. **(E)** Top panel: C11, I11, I12, and O11 of Na$_v$ 1.1 channels during $V_{PSO}$. Bottom panel: C61, I61, I62, and O61 of Na$_v$ 1.6 channels during $V_{PSO}$. **(F)** Voltage response recorded in each node when applying $V_{PSO}$ as a series of voltage clamps at node 10.

## PSO-generated DC waveform drives $V_{PSO}$ and generates onset-free conduction block

Having generated a transmembrane voltage trajectory that caused CSI of Na$_v$ 1.1 and Na$_v$ 1.6 channels, our next step was to design a stimulus waveform to drive this voltage profile in the axon. We again applied PSO, with the particles defined as the electrode currents at each time step, to design an extracellular stimulus to drive $V_{PSO}$ in node 10 using an extracellular monopolar electrode placed 1 mm above the central node (Fig 3A). The goal of the PSO was to minimize the difference between the membrane voltage recorded in node 10 and the optimized voltage profile $V_{PSO}$. The resulting current $I_{PSO}$ increased nonlinearly from 0 mA to a plateau of –0.69 mA (Fig 3B) and drove the transmembrane potential to track $V_{PSO}$. No AP was generated in the axon by $I_{PSO}$ (Fig 3C) and it successfully blocked nerve conduction. When the time delay ($\Delta\tau$) between the onset of an intracellular test pulse and the onset of $I_{PSO}$ was $\leq$ 35 ms, the single spike propagated through the axon (Fig 3D). With $\Delta\tau \geq$ 36 ms, the transmission of single spike was blocked (Fig 3D), as were spike trains at 50 Hz, 100 Hz or 200 Hz (Fig 3E–3G). Thus, the PSO-generated waveform $I_{PSO}$ blocked nerve conduction without activating an onset response in the hybrid model axon.

## KHFAC waveform for onset-free conduction block

The DC $I_{PSO}$ waveform was not charge-balanced, and thus can cause irreversible faradaic reactions at the electrode—electrolyte interface [35], and these may result in electrode and or nerve damage when applied for long periods of time [12, 16, 36]. We therefore used a scaled version of the DC $I_{PSO}$ waveform as the envelope of symmetric rectangular charge-balanced biphasic pulses [12, 37], applied at 10 kHz to block nerve conduction (Fig 4A, bottom). However, there was a time delay after the onset of resulting KHFAC waveform $I_{BI}$ and complete conduction block (Fig 5A), and this delay depended on the amplitude scale factor (Fig 5B). The minimum scale factor for producing conduction block was 1.0, but we used a scale factor of 1.5, which enabled the KHFAC waveform to generate conduction block after a minimum time delay.

The KHFAC waveform $I_{BI}$, delivered using a monopolar electrode placed 1 mm over node 10, generated passive oscillations in transmembrane voltage at node 10, and the upper envelope of the transmembrane voltage response followed the optimized voltage profile $V_{PSO}$ (Fig 4A, top). The intrinsic low-pass filtering and rectification of AC electrical signals by the cell membrane prevented neural firing from following the KHFAC stimulus, while the envelop of the PSO-based waveform $I_{BI}$ drove the Na$_v$ 1.1 channels to CSI (Fig 4B). The fraction C11 of Na$_v$ 1.1 channels in C1 state was higher than that in response to the DC waveform $I_{PSO}$ (Fig 4B), which enabled more channels to enter O1 state instead of I2 and resulted in more Na$^+$ influx through the Na$_v$ 1.1 channels, causing a depolarized peak in the transmembrane voltage of node 10. Importantly, this depolarized peak did not generate a propagating AP (Fig 4C), and the KHFAC waveform enabled onset-free conduction block. When the time delay ($\Delta\tau$) between the onset of an intracellular test pulse and the onset of $I_{BI}$ was $\leq$ 47 ms, the single spike propagated through the axon (Fig 5A). With $\Delta\tau \geq$ 48 ms, the transmission of the single

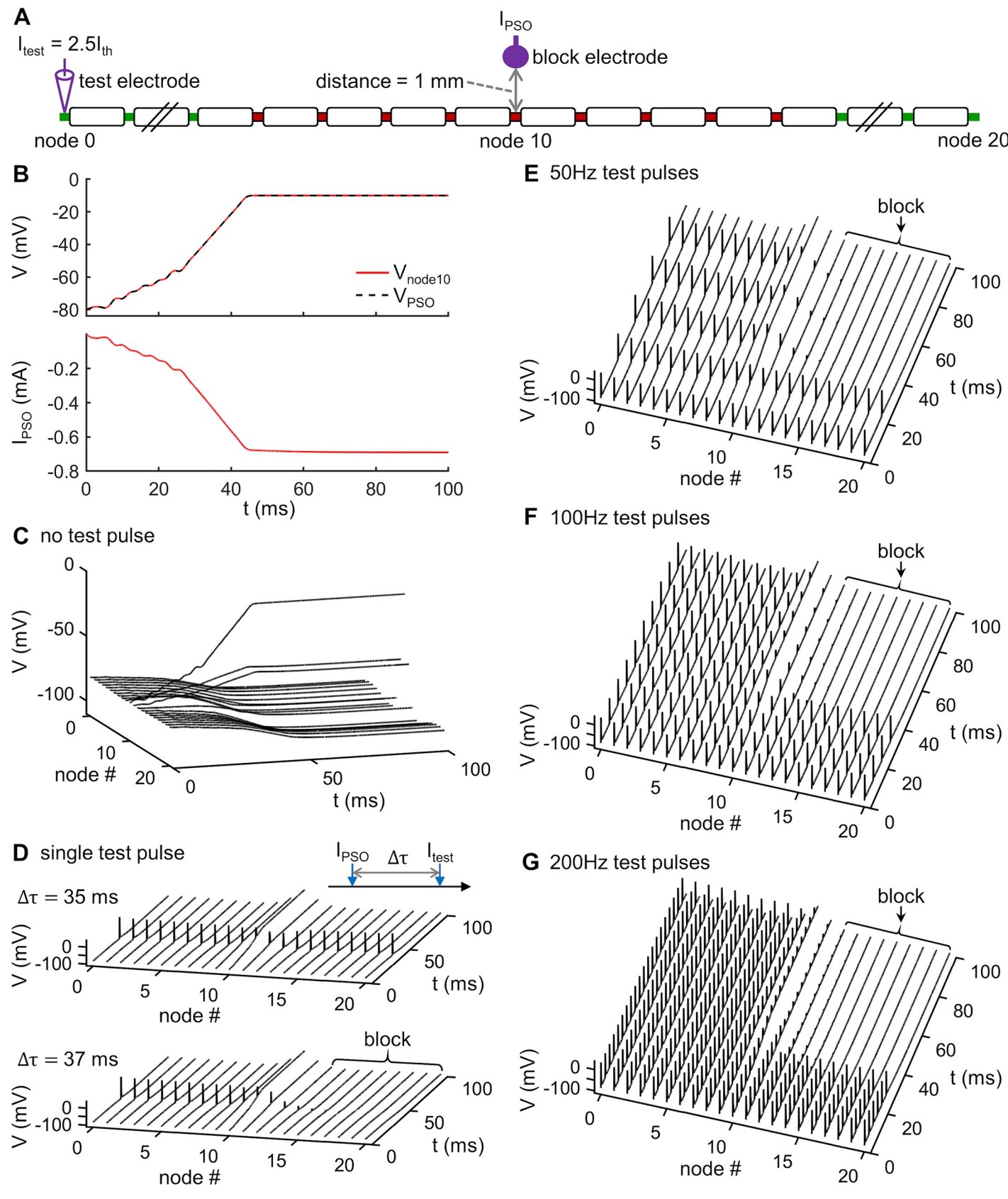

**Fig 3. PSO-generated DC waveform drives $V_{PSO}$ and generates onset-free conduction block.** (A) Simulation setup. A monopolar block electrode was placed 1 mm over node 10, and suprathreshold intracellular test pulses were delivered at node 0 to generate propagating APs. (B) Top panel: Voltage response recorded at node 10 (red solid line) and the PSO-generated voltage profile $V_{PSO}$ (black dotted line). Bottom panel: PSO-generated DC waveform $I_{PSO}$ for driving $V_{PSO}$ in node 10. (C) Voltage response recorded in each node after applying $I_{PSO}$. No test pulse was delivered at node 0. (D) Propagation or block of a single test AP along the axon initiated at a time delay $\Delta\tau$ after the onset of $I_{PSO}$. Top panel: $\Delta\tau$ = 35 ms, bottom panel: $\Delta\tau$ = 37 ms. (E) Block of train of 50 Hz test APs by $I_{PSO}$. (F) Block of train of 100 Hz test APs by $I_{PSO}$. (G) Block of train of 200 Hz test APs by $I_{PSO}$.

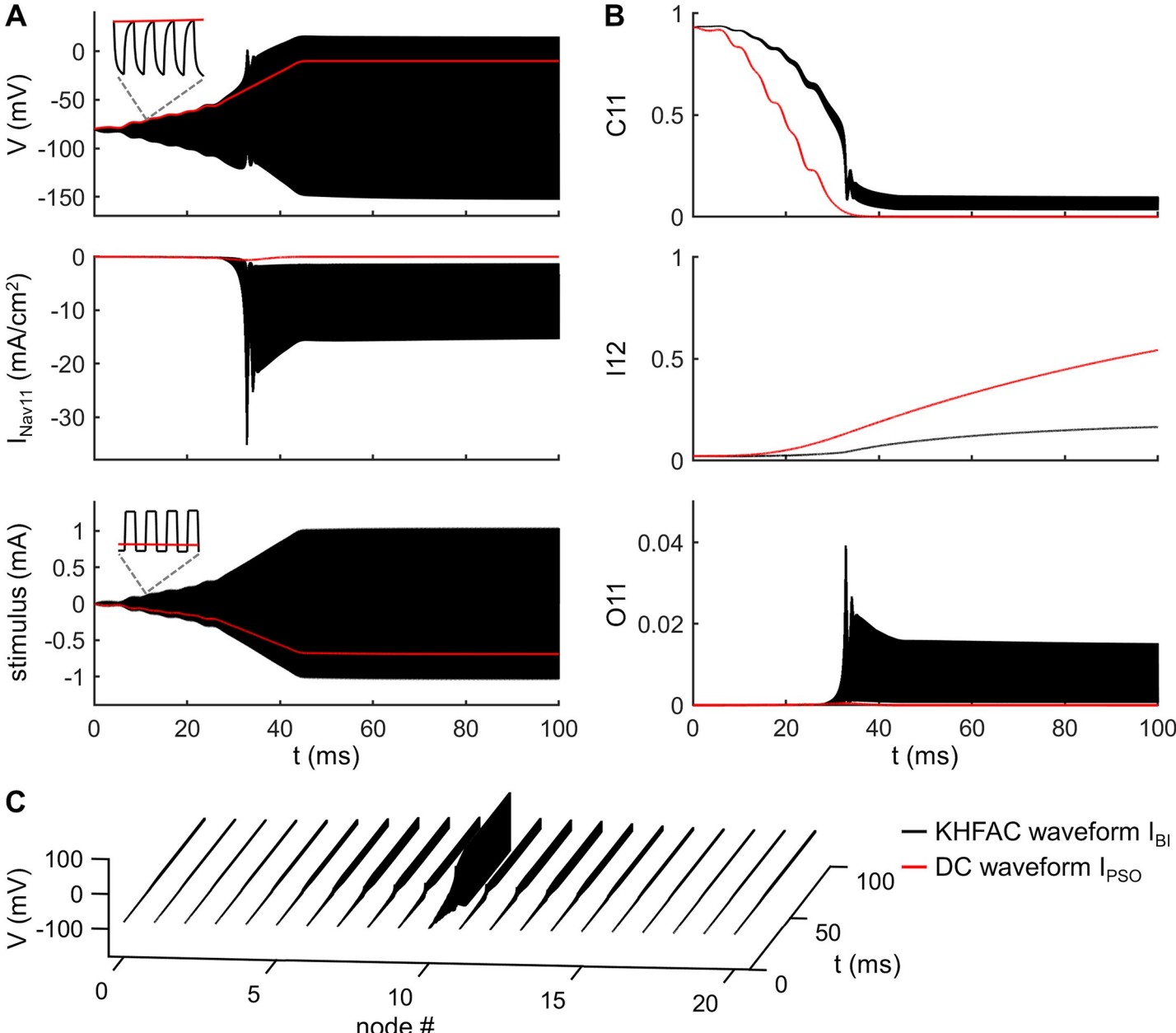

**Fig 4. PSO-based charge-balance biphasic KHFAC waveform for onset-free conduction block.** (A) Transmembrane voltage (top) and $Na_v$ 1.1 current $I_{Nav11}$ (center) recorded in node 10 in response to KHFAC current $I_{BI}$ (bottom). The frequency of $I_{BI}$ was 10 kHz, and each pulse was rectangular, symmetric, and charge-balanced. The envelope of $I_{BI}$ was determined by multiplying $I_{PSO}$ (red line) by a scale factor of 1.5. (B) Fractions of $Na_v$ 1.1 channels in the states C1 (C11), I2 (I12), and O1 (O11). (C) Voltage response recorded in each node after applying $I_{BI}$ with a monopolar electrode placed 1 mm above the central node. No test pulse was injected at node 0. The red lines in (A) and (B) were the responses associated with DC waveform $I_{PSO}$.

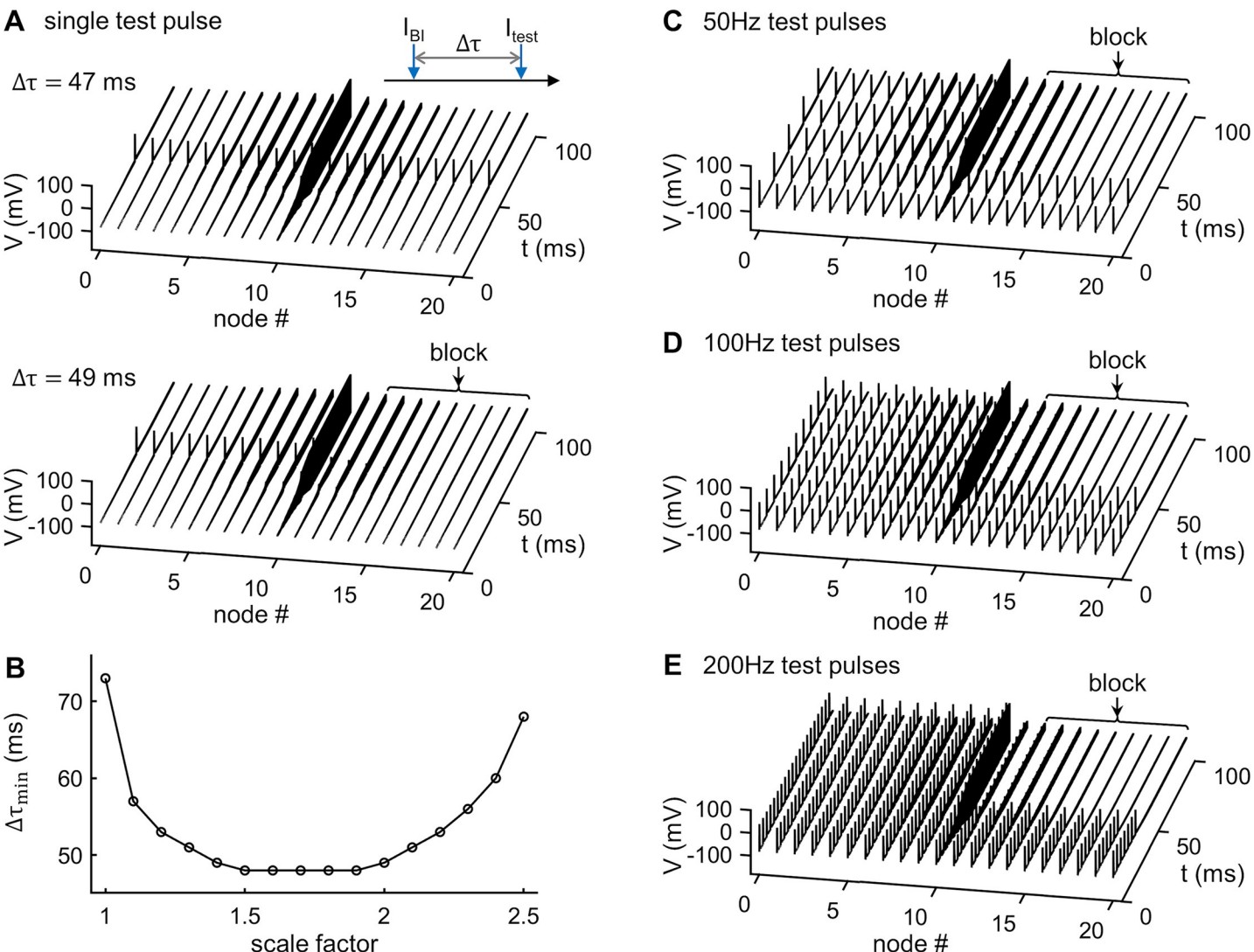

**Fig 5. Onset-free conduction block by PSO-based KHFAC waveform. (A)** Propagation or block of a single test AP along the axon initiated at a time delay $\Delta\tau$ after the onset of $I_{BI}$. Top panel: $\Delta\tau$ = 47 ms, bottom panel: $\Delta\tau$ = 49 ms. A scale factor of 1.5 was used to design the envelope of $I_{BI}$. **(B)** Minimum time delay $\Delta\tau_{min}$ for producing conduction block as a function of scale factor. **(C)** Block of train of 50 Hz test APs by $I_{BI}$. **(D)** Block of train of 100 Hz test APs by $I_{BI}$. **(E)** Block of train of 200 Hz test APs by $I_{BI}$. PSO-based KHFAC waveform $I_{BI}$ was applied using a monopolar block electrode placed over the central node at an electrode-fiber distance of 1 mm.

spike was blocked (Fig 5A), and the minimum delay $\Delta\tau_{min}$ for producing conduction block was dependent on the scale factor (Fig 5B), reaching a minimum when the scale factor was 1.5 ~ 1.9. Further, $I_{BI}$ completely blocked conduction of spike trains at 50 Hz, 100 Hz, or 200 Hz (Fig 5C–5E). Thus, the PSO-based charged-balanced KHFAC waveform blocked nerve conduction without generating an onset response.

## Discussion

We implemented a novel design process to develop a KHFAC stimulus waveform to enable nerve conduction block without the onset response that occurs with conventional KHFAC waveforms. We implemented Markov-type kinetic models to simulate CSI of $Na_v$ 1.1 and $Na_v$ 1.6 channels in a novel hybrid model of a mammalian myelinated nerve fiber. We used engineering optimization with a PSO algorithm first to design an optimized transmembrane

voltage trajectory to drive the two VGSCs to CSI without first causing activation, and subsequently, to generate an extracellular DC stimulation waveform that drove the voltage profile in the central node of the model to follow the optimized trajectory. Finally, we designed a charge-balanced 10 kHz biphasic waveform based on the PSO-generated DC waveform, and the PSO-based KHFAC waveform produced robust conduction block without an onset response.

In *in vivo* experiments, a KHFAC waveform is commonly delivered using an extracellular electrode and can affect a number of axons with different electrode-fiber distances. We calculated the PSO-based biphasic waveform $I_{BI}$ in a 21-node model at distance of 1 mm, and this waveform did not produce onset firing. The block threshold increases with increasing electrode-fiber distance [17, 38, 39], and we examined whether our PSO-based KHFAC waveform optimized at a specific location produced onset-free conduction block in the axon at other locations. Applying $I_{BI}$ with same plateau amplitude generated an onset response in the model at an electrode-fiber distance of 0.3 mm (S2 Fig). These closer nerve fibers had a block threshold lower than the plateau amplitude of $I_{BI}$, and as the KHFAC amplitude approached the plateau, the node underneath the blocking electrode was strongly depolarized and APs were initiated in adjacent nodes. While the PSO-based biphasic waveform eliminated the onset response in nerve fibers at certain locations, it may still activate onset firing at the other locations.

In *in vivo* experiments, a KHFAC waveform delivered using an extracellular electrode can also affect axons with different fiber diameters. The block threshold decreases with increasing fiber diameter [17, 38, 39], and we examined whether the PSO-based KHFAC waveform produced onset-free conduction block in smaller diameter (2 μm and 8.7 μm) model axons. The internodal lengths of 2 μm or 8.7 μm axons are much shorter than those of the 10 μm fiber (S1 Table), which significantly reduced the total length of the 21-node model axon. Since the electrode-fiber distance was a fixed value (1 mm), we increased the number of nodes to 251 to examine conduction block using the PSO-based KHFAC waveform and to avoid potential end effects, as the ends of the 21-node model axons with shorter internodal lengths were much closer to the electrode. When scaling the plateau amplitude of $I_{BI}$ to the block threshold of a 251-node model at a diameter of 2 μm, two APs were generated as the KHFAC amplitude approached the block threshold (S3 Fig). Applying $I_{BI}$ with same plateau amplitude to a 251-node model at a diameter of 8.7 μm also produced an onset response. While the PSO-based KHFAC waveform eliminated onset response in the fibers for which it was designed, it may generate onset responses in other fiber diameters.

The original MRG models used the HH-type formalism to describe fast $Na^+$ and persistent $Na^+$ channels in each node [31], and we designed the waveform using the hybrid model incorporating Markov-type channels. Applying our PSO-based biphasic waveform to a 21-node 10 μm HH-type MRG axon, i.e., without the Markov-type VGSCs in the central nodes, generated eight APs as KHFAC amplitude approached the block threshold (S4 Fig). The initial ramp of biphasic stimulus drove the central node under the block electrode to suprathreshold depolarization, which enabled the initiation of onset firing. The VGSC gating variables $m$ and $h$ operate independently of each other, and no transition occurred between them as KHFAC amplitude increased. In contrast, the Markov-type $Na_v$ 1.1 channels moved from the closed state to inactivation without fully opening in response to the PSO-based biphasic waveform, and thus no onset response was generated. To reduce overall computational demands, we implemented $Na_v$ 1.1 and $Na_v$ 1.6 channels only in the central nine nodes of a 21-node MRG model, but the specific number of Markov-type nodes did not alter the block threshold or the lack of an onset response in the hybrid model (S5 Fig).

An earlier study of somatic voltage clamp [40] reported that applying a voltage pre-pulse inactivated axonal channels after evoking an axonal spike, and the voltage steps following the

pulse no longer triggered axonal spikes. Our study showed that directly applying voltage clamp of the PSO-generated voltage profile $V_{PSO}$ at the central node generated CSI of VGSCs, and no APs were escaped from the node. Further, we used a monopolar point source electrode to design PSO-based DC and KHFAC waveforms in the hybrid MRG model, while bipolar and tripolar cuff electrodes are commonly used *in vivo* [6, 14, 15, 17]. Linearly ramping KHFAC amplitude from zero to block threshold in rat sciatic nerve did not eliminate the onset response with a tripolar electrode [14], while linearly ramping KHFAC waveforms from non-zero amplitudes reduced the onset response with a bipolar electrode [15]. We re-calculated the PSO-generated DC and biphasic waveforms using representations of bipolar and tripolar cuff electrodes, and the resulting KHFAC waveform was similar for each of three electrodes. Using a bipolar cuff, there was onset firing activated in the 21-node hybrid model as the KHFAC amplitude approached the block threshold (S6 Fig), but using the tripolar cuff, no onset response was generated by the initial ramp of the KHFAC waveform, as observed with the original monopolar electrode.

Other waveforms were proposed to eliminate the onset response. Tai *et al.* [41] predicted that ramping KHFAC amplitude from 0 mA to a level sufficient to produce block eliminated the onset response in a HH model of unmyelinated giant squid axons. Miles *et al.* [14] showed that the KHFAC ramp that started from zero amplitude did not eliminate the onset response in either the HH-type MRG model or in experiments on rat sciatic nerve. Vrabec *et al.* [15] recently reported that ramping KHFAC amplitudes up to 125% of block threshold from non-zero amplitudes were successful in reducing the onset response in rat sciatic nerve. *In vivo* recordings demonstrated that the onset response was minimized with high amplitudes and high frequencies [9, 17, 42], and Gerges *et al* [13] showed that a transition from a high amplitude and high frequency to a low amplitude and low frequency during KHFAC nerve block minimized or eliminated the onset response with longer transition times. A DC block was also introduced to eliminate the onset response. Ackermann *et al.* [43] reported that a brief DC block in combination with KHFAC prevented the onset firing, but such DC was found to produce nerve damage and prolonged conduction failure. Subsequently, Franke *et al.* [12] suggested that combining KHFAC and charge-balanced DC block significantly reduced or completely prevented the onset response without a change in nerve conduction by using a block electrode of high capacitance materials. The PSO-generated DC waveform also eliminated the onset response in the hybrid MRG axon, but our DC block was not charge-balanced.

We presented an engineering optimization method to design a charge-balanced biphasic current waveform that eliminated the onset response during KHFAC nerve block. Our PSO-based KHFAC waveform completely prevented the onset firing in axons at specific locations and with specific diameters, which suggested that it is possible to achieve KHFAC nerve block without an onset response by moving VGSCs to CSI.

## Methods

### Axon model

We used the MRG double-cable myelinated axon model to simulate KHFAC conduction block, and this model was originally developed to reproduce a wide range of excitation properties of mammalian peripheral nerve fibers [31]. The model was composed of equivalent electrical circuits for both nodal and internodal segments (Fig 1A). The nodal circuit included fast Na$^+$ ($I_{Naf}$), persistent Na$^+$ ($I_{Nap}$), slow K$^+$ ($I_{KS}$), and linear leakage ($I_L$) currents, and the membrane capacitance $C_n$, and the ionic currents were originally modeled by using HH-type kinetics. $I_{Naf}$ was responsible for AP initiation in each node (maximum conductance: $\bar{g}_{Naf} = 3$ S/cm$^2$), and $I_{Nap}$ was responsible for generating a non-inactivating current at the afterpotentials

(maximum conductance: $\bar{g}_{\mathrm{Nap}} = 0.01$ S/cm$^2$). The resting potential in each node was –80 mV. The axon-specific parameters were implemented following McIntyre *et al* [31, 44] (S1 and S2 Tables) and were previously validated to be consistent with experimental observations of KHFAC conduction block [38, 39, 45]. The axon models were solved in the NEURON simulation environment [46] (version 7.7) using a time step of 0.005 ms.

## Markov-type models of VGSCs

There are two types of models commonly used to reproduce the electrophysiological behaviors of VGSCs, i.e., HH- and Markov-type models. The HH-type models represent an ionic channel as an assembly of several independent gating particles, which are not sufficient to capture specific kinetics of VGSCs [47, 48]. Unlike the HH formalism, Markov-type models represent an ion channel as a series of conformational states of the channel protein and a collection of transitions between them [18, 22], which allowed us to reproduce the dependence of Na$^+$ inactivation on activation. We used Markov-type kinetic models of VGSCs to design the transmembrane voltage trajectory for causing Na$^+$ CSI.

The Markov-type models used in our simulations were developed by Balbi *et al* [30] and were validated against electrophysiological data from all VGSC isoforms. The model framework was detailed, unifying, and computationally efficient, and can account for different features of each human VGSC isoform with a minimal set of states and transitions. The model included six states, and the state diagram and transitions of the Markov-type channels are shown in Fig 1B. C1 and C2 were two closed states, O1 and O2 were two open states, and I1 and I2 were two inactivation states. The transition from O1 to I1 was irreversible, and the transitions between the other consecutive states were reversible. The dynamics of fraction of VGSCs in each state were governed by following Markovian equations [30]

$$dC1/dt = A_{I1C1}I1 + A_{C2C1}C2 - (A_{C1C2} + A_{C1I1})C1 \tag{1}$$

$$dC2/dt = A_{C1C2}C1 + A_{O1C2}O1 + A_{O2C2}O2 - (A_{C2C1} + A_{C2O1} + A_{C2O2})C2 \tag{2}$$

$$dO1/dt = A_{C2O1}C2 + A_{I1O1}I1 - (A_{O1C2} + A_{O1I1})O1 \tag{3}$$

$$dO2/dt = A_{C2O2}C2 - A_{O2C2}O2 \tag{4}$$

$$dI1/dt = A_{I2I1}I2 + A_{C1I1}C1 + A_{O1I1}O1 - (A_{I1C1} + A_{I1I2} + A_{I1O1})I1 \tag{5}$$

$$dI2/dt = A_{I1I2}I1 - A_{I2I1}I2 \tag{6}$$

Here $A_{w1w2}$ was the transition rate from state $w1$ to state $w2$, which was computed by

$$A_{w1w2} = B_{hyp}^{w1w2}\left[1 + exp\left(\frac{V - V_{hyp}^{w1w2}}{k_{hyp}^{w1w2}}\right)\right]^{-1} + B_{dep}^{w1w2}\left[1 + exp\left(\frac{V - V_{dep}^{w1w2}}{k_{dep}^{w1w2}}\right)\right]^{-1} \tag{7}$$

The Na$^+$ conductance was determined by the fraction of open channels, and the Na$^+$ current was computed by

$$I_{\mathrm{Nav}} = \bar{g}_{\mathrm{Nav}}(O1 + O2)(V - E_{\mathrm{Na}}) \tag{8}$$

where $\bar{g}_{\mathrm{Nav}}$ was the maximum conductance, $V$ was local membrane voltage, and $E_{\mathrm{Na}} = 50$ mV was the Na$^+$ reversal potential.

## Implementation of Markov-type VGSCs

KHFAC waveforms are applied in practice using comparatively small electrodes and generate a locally-acting conduction block in peripheral nerves. In our simulations, there were 21 nodes in the model axon and the simulated block electrode was placed above the central node (i.e., node 10). To reduce overall computational demands, we implemented Markov-type VGSCs in the central nine nodes, which were closest to the block electrode. The other nodes far away from the site of block were modeled with HH-type VGSCs. Note that the specific number of Markov-type nodes did not alter the block threshold or lack of onset response in the hybrid model (S5 Fig).

Experimental data indicated that the Na$_v$ 1.1 and Na$_v$ 1.6 channels are preferentially expressed in the nodes of mammalian peripheral nerve fibers [49–51]. We substituted the HH-type models of $I_{Naf}$ and $I_{Nap}$ in node 6 to node 14 with the Markov-type models of Na$_v$ 1.1 and Na$_v$ 1.6 channels, respectively. The parameters of each transition rate for two VGSCs are provided in S3 Table. We modified only the maximum conductances of $I_{Nav11}$ and $I_{Nav16}$ and did not change any parameter of their transition rates between consecutive states. The maximum conductance of Na$_v$ 1.6 channels was set to $\bar{g}_{Nav16} = 0.01$ S/cm$^2$, which was identical to that of $I_{Nap}$. For Na$_v$ 1.1 channels, if we set $\bar{g}_{Nav11}$ to the minimal value (4.7 S/cm$^2$) for faithfully propagating a single spike from node 0 to node 20, there was substantial attenuation in the AP as it propagated from HH-type nodes to Markov-type nodes (S1 Fig). In this case, the $\bar{g}_{Nav11}$ was too small to propagate APs at high frequencies. For example, when firing rate was 200 spikes per second, APs were lost as they propagated along the axon. We set $\bar{g}_{Nav11}$ to the minimal value (11.9 S/cm$^2$) to propagate faithfully spike trains at rates of up to 400 spikes per second. At $\bar{g}_{Nav11} = 11.9$ S/cm$^2$, there was no attenuation in the APs as they propagated along the axon (S1 Fig). Although $\bar{g}_{Nav11}$ was larger than $\bar{g}_{Naf}$, $I_{Nav11}$ and $I_{Naf}$ had similar magnitude underlying an AP.

## Particle swarm optimization

We applied PSO algorithms to design the transmembrane voltage trajectory for causing Na$^+$ CSI and to generate current waveform to drive this voltage trajectroy, which were implemented in Matlab (version R2016a). PSO is a meta-heuristic optimization method inspired by the information circulation and social behavior of swarms [32]. It uses a population-based search stratergy to solve nonlinear optimization problems, based on cooperation and competition among the particles of a swarm. In PSO, each particle is a point in the search space, which represents a candidate solution of the optimization problem. The movement of a particle in the search space is driven by its own fitness as well as the highest fitness in the swarm. The algorithm iteratively updates the velocity of each particle towards the position of the highest fitness. Through communication between particles over multiple iterations, the particles explore the problem space to identify the optimal solution of the problem.

Mathematically, the position of particle $i$ ($1 \leq i \leq$ M) at the $k$th iteration was described by a vector $\vec{X}_i(k) = [x_{i,1}(k), x_{i,2}(k), \ldots, x_{i,D}(k)]$, where M was the number of particles and D was the dimension of the search space. Each dimension of the particle was bounded between $x_{\min}$ and $x_{\max}$. The personal best position prevoiusly found by particle $i$ was $\vec{P}_i = [p_{i,1}, p_{i,2}, \ldots, p_{i,D}]$, and the best position expericed by all particles in the whole swarm was indicated by $\vec{G} = [g_1, g_2, \ldots, g_D]$. The velocity of particle $i$ in the $k$th iteration was $\vec{V}_i(t) = [v_{i,1}(k), v_{i,2}(k), \ldots, v_{i,D}(k)]$, which determined the movement direction of the particle through the search space. At the $k$th iteration, the $d$-dimension ($1 \leq d \leq$ D) of particle $i$ was updated based on following equations

$$v_{i,d}(k+1) = c\{r_1[p_{i,d} - x_{i,d}(k)] + r_2[g_d - x_{i,d}(k)]\} \tag{9}$$

$$x_{i,d}(k + 1) = x_{i,d}(k) + v_{i,d}(k + 1) \tag{10}$$

where $c = 1.5$ was a learning coefficient, and $r_1$ and $r_2$ were uniform random numbers within [0, 1] used to help promote exploration.

## Implementation of PSO

We applied PSO to generate an optimized voltage trajectory $V_{PSO}$ for causing the CSI of $Na_v$ 1.1 and $Na_v$ 1.6 channels in the 21-node model (Fig 6A). We defined a population of 50 particles as the membrane voltages used to control a time series of voltage clamps at node 10. We recorded the $Na^+$ currents through $Na_v$ 1.1 and $Na_v$ 1.6 channels as well as their fractions in states C1, I1, I2, and O1 during voltage clamp. We defined a multi-objective optimization problem: find a voltage trajectory to maximize the fractions of $Na_v$ 1.1 and $Na_v$ 1.6 channels in inactivated states while minimizing the fractions of the two VGSCs in open states. Mathematically, the following cost function $f_1$ was defined to evaluate each solution,

$$\text{Minimize} \quad f_1 = \sum_{j=1}^{N} [B(O11_j + O12_j + O61_j + O62_j) - (I12_j + I62_j)] \tag{11}$$

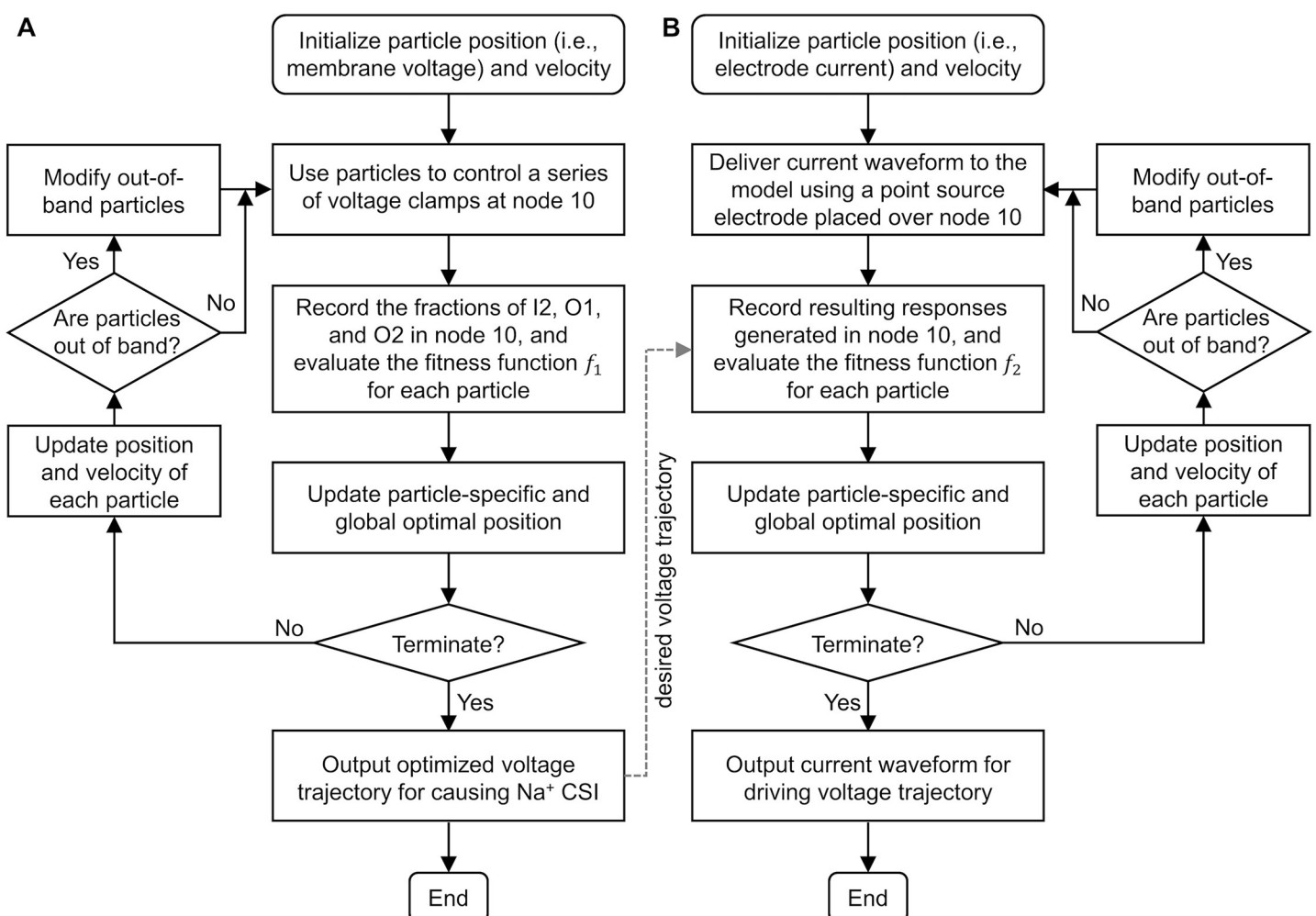

**Fig 6.** Flowchart of the PSO algorithm for **(A)** designing a transmembrane voltage trajectory for causing $Na^+$ CSI and **(B)** generating DC waveform for driving that trajectory.

where $O11$, $O12$, and $I12$ were the fractions of $Na_v$ 1.1 channels in states O1, O2 and I2, $O61$, $O62$, and $I62$ were the fractions of $Na_v$ 1.6 channels in states O1, O2 and I2, $j$ corresponded to the time point, and $B = 20000$ was a weight used to reduce the fractions in open states. The lower bound of each particle was $x_{min} = -80$ mV (i.e., resting potential), and the upper bound was $x_{max} = -10$ mV. The time constant of inactivation state I1 was smaller than I2, and the optimization results were sensitive to the changes in I1. Thus, we excluded the fraction of state I1 from the fitness function $f_1$. The initial positions of 50 particles were defined to be randomly distributed around a straight line, which linearly increased from $x_{min}$ to $x_{max}$.

Subsequently, we used PSO to generate a current waveform for driving the optimized voltage trajectory in the central node of the 21-node model (Fig 6B). We placed a point source electrode over node 10 at an electrode-fiber distance of 1 mm. We defined 50 particles as a time series of electrode currents. The extracellular medium was assumed to be infinite homogenous and isotropic with a conductivity of $\sigma = 0.303$ S/m [52, 53]. For a segment at location $(x, y, z)$, the extracellular potential $V_{ext}$ was calculated as [53, 54]

$$V_{ext}(x, y, z) = \frac{I_{ext}}{4\pi\sigma\sqrt{(x - x_0)^2 + (y - y_0)^2 + (z - z_0)^2}} \qquad (12)$$

where $I_{ext}$ was the amplitude of electrode current, and $(x_0, y_0, z_0)$ was the location of the point source electrode. We recorded the resulting transmembrane voltage in node 10 at each iteration. We set $V_{PSO}$ as the desired voltage trajectory, and defined an one-objective optimization problem: find electrode currents to minimize the difference between transmembrane voltage $V_{node10}$ recorded in node 10 and desired voltage trajectory $V_{PSO}$. Mathematically, the following cost function $f_2$ was defined to evaluate the quality of this problem solution

$$\text{Minimize} \quad f_2 = \sum_{j=1}^{N}(V_{node10,j} - V_{PSO,j})^2 \qquad (13)$$

where $j$ corresponded to the time point. By using the point source electrode, cathodic stimulus depolarized node10 and anodic stimulus hyperpolarized node10. Since $V_{PSO}$ started from the resting potential, the upper bound of each particle was $x_{max} = 0$ mA. The lower bound was defined to $x_{min} = -0.7$ mA, which was sufficient to drive transmembrane voltage in node 10 to a plateau level higher than $-10$ mV. The initial positions of particles were defined to be randomly distributed around a straight line from the $x_{min}$ to $x_{max}$.

When solving the optimal solutions for $f_1$ and $f_2$, the time step of PSO was 2 ms and the search space dimension was D = 50. The termination criterion was that the algorithm generation reached a limit of 1000, which was sufficient for convergence. When designing the optimized voltage trajectory $V_{PSO}$ for causing $Na^+$ CSI, the minimum $f_1$ converged to within 1% of the final iteration by 330 iterations (S7 Fig). When generating the DC waveform $I_{PSO}$ for driving the optimized trajectory $V_{PSO}$, the minimum fitness $f_2$ after 499 iterations converged to within 1% of the final iteration (S8 Fig).

## Simulation protocols for KHFAC conduction block

After generating PSO-based DC and KHFAC waveforms for conduction block, we first examined whether they activated onset responses. We used an extracellular point source electrode to deliver blocking currents to the central node (10) of the model, and the electrode-fiber distance was 1 mm. Subsequently, we determined whether the PSO-generated waveform blocked nerve conduction. We applied an intracellular monopolar electrode to deliver suprathreshold test pulses (width: 0.1 ms and amplitude: $2.5I_{th}$) at one end of the axon (node 0), which was used to stimulate APs in node 0 after the onset of blocking currents and test whether these APs

propagated through the site of the block electrode. $I_{th}$ = 0.43 mA was the threshold of intracellular test pulse for activation of an AP in node 0.

A 251-node model was used to examine the conduction block at the fiber diameters (2 μm and 8.7 μm) less than 10 μm using PSO-based KHFAC waveform. Each fiber diameter had specific lengths for all segments (S1 Table), which were identified based on experimental data. The Markov-type Na$_v$ 1.1 and Na$_v$ 1.6 channels were implemented in node 121 to node 129, and the block electrode was placed 1 mm above node 125.

## Supporting information

**S1 Fig. Setting maximum conductance of Na$_v$ 1.1 channels.** Propagation of a test AP initiated in node 0 along the 21-node and 10 μm model at **(A)** $\bar{g}_{Nav11}$ = 4.7 S/cm$^2$ and **(B)** 11.9 S/cm$^2$. $\bar{g}_{Nav11}$ = 4.7 S/cm$^2$ was the minimum conductance for faithfully propagating a single AP along the axon, and $\bar{g}_{Nav11}$ = 11.9 S/cm$^2$ was the minimum conductance for faithfully propagating spike trains at rates of up to 400 Hz along the axon. $I_{Naf}$ was recorded in node 3, and $I_{Nav11}$ was recorded in node 10. Left panels: a single test pulse (width: 0.1 ms and amplitude: 2.5$I_{th}$) was delivered at the node 0. Right panels: 200 Hz test pulses were delivered at the node 0. Maximum conductance of HH-type $I_{Naf}$ was $\bar{g}_{Naf}$ = 3.0 S/cm$^2$, maximum conductance of HH-type $I_{Nap}$ was $\bar{g}_{Nap}$ = 0.01 S/cm$^2$, and maximum conductance of Markov-type $I_{Nav16}$ was $\bar{g}_{Nav16}$ = 0.01 S/cm$^2$.
(TIF)

**S2 Fig. Conduction block at electrode-fiber distance of 0.3 mm by PSO-based KHFAC waveform. (A)** Simulation setup. A monopolar block electrode was placed 0.3 mm over the central node of 21-node 10 μm diameter model nerve fiber. An intracellular test pulse (width: 0.1 ms and amplitude: 2.5$I_{th}$) was delivered at node 0 to generate a propagating AP at $t$ = 120 ms. **(B)** Transmembrane voltages (top) recorded in node 0 and node 20 in response to KHFAC waveform $I_{BI}$ (bottom). A scale factor of 1.5 was used to design the envelope of $I_{BI}$, and the green dotted line was the block threshold at electrode-fiber distance of 0.3 mm. **(C)** Voltage responses recorded in node 10 to node 14.
(TIF)

**S3 Fig. Effects of fiber diameter on conduction block by PSO-based KHFAC waveform. (A)** Simulation setup. A monopolar block electrode was placed 1 mm over node 125 and delivered KHFAC waveform $I_{BI}$ to a 251-node model. An intracellular test pulse (width: 0.1 ms and amplitude: 2.5$I_{th}$) was delivered at node 0 to generate a propagating AP at $t$ = 120 ms. Markov-type Na$_v$ 1.1 and Na$_v$ 1.6 channels were implemented in node 121 to node 129. **(B)** Block of a test AP along the axon by $I_{BI}$ with a fiber diameter of 2.0 μm. **(C)** Block of a test AP along the axon by $I_{BI}$ with a fiber diameter of 8.7 μm. In **(B)** and **(C)**, the green dotted lines were the block threshold at each fiber diameter.
(TIF)

**S4 Fig. Conduction block in HH-type model by PSO-based KHFAC waveform. (A)** Block of a test AP along the 21-node 10 μm diameter model nerve fiber by PSO-based waveform $I_{BI}$. No Markov-type VGSCs were implemented in the central nodes. The block electrode was placed 1 mm above node 10, and a single test pulse (width: 0.1 ms and amplitude: 2.5$I_{th}$) was delivered at node 0 to generate a propagating AP at $t$ = 70 ms. **(B)** Top panel: transmembrane voltage recorded in node 10 in response to KHFAC waveform $I_{BI}$. Center panel: activation gating variable $m$ and inactivation gating variable $h$ of fast Na$^+$ current. Bottom panel: PSO-based KHFAC waveform $I_{BI}$ with a plateau amplitude scaled to the block threshold.
(TIF)

**S5 Fig. Effects of number of Markov-type nodes on conduction block by PSO-based KHFAC waveform. (A)** Simulation setup. We implemented $Na_v$ 1.1 and $Na_v$ 1.6 channels in node (10−n) to node (10+n), where $0 \leq n \leq 10$. A monopolar block electrode was placed 1 mm over node 10, and an intracellular test pulse (width: 0.1 ms and amplitude: $2.5I_{th}$) was delivered at node 0 to generate a propagating AP at $t$ = 120 ms. **(B)** Block threshold as a function of the number of Markov-type nodes. At $3 \leq n \leq 10$, PSO-based KHFAC waveform $I_{BI}$ produced onset-free conduction block. At $0 \leq n \leq 2$, no conduction block occurred and onset firing was activated by $I_{BI}$. **(C)** Propagation of a test AP along the axon by $I_{BI}$ (n = 1). A scale factor of 1.0 was used to design the envelope of $I_{BI}$, and the number of Markov-type nodes was three.
(TIF)

**S6 Fig. PSO-based KHFAC waveform designed using bipolar and tripolar cuff electrodes. (A)** We interpolated the extracellular potential at each segment of 21-node hybrid model from a finite element model (FEM) of the rat tibial nerve, which had a bipolar nerve cuff and a model nerve fiber that was 0.398 mm away from the electrode. Since the bipolar cuff resulted in the largest depolarization in node 11, we applied PSO algorithm to generate a DC waveform $I_{PSO}$ to drive the optimized voltage trajectory $V_{PSO}$ in node 11. The envelop of KHFAC waveform $I_{BI}$ was determined by multiplying $I_{PSO}$ by a scale factor of 0.73, and the plateau amplitude of $I_{BI}$ was the block threshold. The resulting KHFAC waveform blocked nerve conduction with activation of an onset response. **(B)** We used a FEM of a tripolar cuff on the rat tibial nerve to interpolate the extracellular potentials of 21-node hybrid model, and the electrode-fiber distance was 0.398 mm. The DC waveform $I_{PSO}$ was designed to drive $V_{PSO}$ in node 10. A scale factor of 0.93 was applied to scale the plateau amplitude of $I_{BI}$ to the block threshold. The resulting KHFAC waveform produced onset-free conduction block. A single test pulse (width: 0.1 ms and amplitude: $2.5I_{th}$) was injected in node 0 at $t$ = 70 ms. Fiber diameter was 10 μm.
(TIF)

**S7 Fig. Progression of PSO for generating transmembrane voltage trajectory to cause $Na^+$ CSI. (A)** Changes of transmembrane voltage trajectory across iterations. The sequence of plots show the voltage profile with the minimum cost $f_1$ at each indicated iteration. **(B)** Minimum cost $f_1$ of 50 particles at each iteration. Red dotted line is 1.01 times the minimum cost at the final generation. **(C)** Voltage response recorded in each node when applying the voltage profile at each indicated iteration as a series of voltage clamps at node 10. Fiber diameter was 10 μm.
(TIF)

**S8 Fig. Minimum cost function $f_2$ of 50 particles at each iteration.** Red dotted line was 1.01 times the minimum cost at the final generation.
(TIF)

**S1 Table. Geometric parameters of MRG models**
(DOCX)

**S2 Table. Electrical parameters of MRG models**
(DOCX)

**S3 Table. Parameters of the transition rates for $Na_v$ 1.1 and $Na_v$ 1.6 channels**
(DOCX)

## Acknowledgments

We would like to thank Edgar Peña, Ph.D., for comments on designing PSO-based KHFAC waveforms and Nathan Titus for implementing the PSO algorithm on the Duke Shared Compute Cluster.

## Author Contributions

**Conceptualization:** Warren M. Grill.

**Data curation:** Guosheng Yi.

**Formal analysis:** Guosheng Yi.

**Funding acquisition:** Guosheng Yi, Warren M. Grill.

**Methodology:** Guosheng Yi, Warren M. Grill.

**Project administration:** Warren M. Grill.

**Supervision:** Warren M. Grill.

**Visualization:** Guosheng Yi, Warren M. Grill.

**Writing – original draft:** Guosheng Yi.

**Writing – review & editing:** Guosheng Yi, Warren M. Grill.

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
