## [Decision Letter · Decision Letter 0]

4 Jan 2020

Dear Dr Grill,

Thank you very much for submitting your manuscript, 'Kilohertz waveforms optimized to produce closed-state Na+ channel inactivation eliminate onset response in nerve conduction block', to PLOS Computational Biology. As with all papers submitted to the journal, yours was fully evaluated by the PLOS Computational Biology editorial team, and in this case, by independent peer reviewers. The reviewers appreciated the attention to an important topic but identified some aspects of the manuscript that should be improved.  Indeed, you will see that in some cases the reviewers ask for clarification or more details, and they have also raised a variety of open-ended questions, which reflect their interest in the work.  We hope that you will find these useful and thought-provoking and look forward to reading your responses.

We would therefore like to ask you to modify the manuscript according to the review recommendations before we can consider your manuscript for acceptance. Your revisions should address the specific points made by each reviewer and we encourage you to respond to particular issues.  Please note while forming your response, if your article is accepted, you may have the opportunity to make the peer review history publicly available. The record will include editor decision letters (with reviews) and your responses to reviewer comments. If eligible, we will contact you to opt in or out.

- Supporting Information uploaded as separate files, titled 'Dataset', 'Figure', 'Table', 'Text', 'Protocol', 'Audio', or 'Video'.

We hope to receive your revised manuscript within the next 30 days. If you anticipate any delay in its return, we ask that you let us know the expected resubmission date by email at ploscompbiol@plos.org.

Sincerely,

Jonathan Rubin

Associate Editor

PLOS Computational Biology

Kim Blackwell

Deputy Editor

PLOS Computational Biology

[LINK]

Reviewer's Responses to Questions

**Comments to the Authors:**

Reviewer #1: An interesting, drug-free method for blocking nerve conduction relies on inactivation of voltage-gated sodium (Nav) channels via externally applied, high-frequency electrical stimuli. This technique can potentially be used to treat pain and spasticity. However, one caveat is that the nerve responds to the onset of the stimulus, which can cause transient muscle contractions and pain.

Yi and Grill propose here a procedure for optimizing the stimulus waveform in such a way that the onset response is eliminated. To generate action potentials, Nav channels first activate, open and generate current, then inactivate and remain inactivated for a period of time that makes the nerve refractory to stimuli. However, Nav channels can also inactivate without opening, a process termed closed state inactivation (CSI). The kHz waveform designed by Yi and Grill exploits CSI.

The problem addressed is important and interesting, the modeling work is good, and the manuscript is clear and well written. Overall, I really enjoyed reading the manuscript and I didn't see any major issue. I only have a number of minor comments and suggestions.

Fig. 2: I think it might be good to show what happens when a standard voltage step is applied, for readers who are less familiar with Nav channels, and also to put the magnitude of the residual INa in context. Otherwise, the current graph in A is uninterpretable.

The states could be directly called C11, I11, etc., instead of SC11, SI11, etc.

Do I understand correctly that the voltage was clamped only at one node (10), and the rest of the axon was completely free? If so, I am still a little surprised that no APs were generated (escaped) farther away from node 10. I think it's really important to show a waveform that is not fully optimized and permits APs to escape, side by side with the optimized waveform, so readers understand what has been achieved here. The authors might want to cite a paper where this phenomenon of escaped APs is actually exploited to selectively silence axonal Nav channels (Milescu at al, 2010).

The optimal waveform was obtained here in the absence of an external stimulus, but would it be different if excitation were present?

The authors might want to emphasize/explain that the goal was not to block conduction in the entire axon by CSI, but only at one node (or a few).

The study convinced me that, in general, a waveform can be designed to produce conduction block without onset response. The waveform discussed here works with a computational compartmental model that describes a certain type of axon, and the authors admit that it may not work with other types. Is there a strategy that would allow a clinical device to generate an optimal waveform for a specific patient? That is, without triggerring an onset response during the optimization process, or at least reduce that. For example, wouldn't it be safe to just ramp the voltage up so very slowly, in hundreds of ms, say? Could you show a simulation with a much shallower ramp? If the optimization was started with a square waveform, I could see how the optimizer would stop when it was just shallow enough. Of course, I could be entirely wrong, and this waveform could really be the only one that works. Please explain and illustrate.

Could you give estimates on how long would optimization take on a desktop computer?

Reviewer #2: review is uploaded

**Have all data underlying the figures and results presented in the manuscript been provided?**

Reviewer #1: Yes

Reviewer #2: Yes

PLOS authors have the option to publish the peer review history of their article (what does this mean?). If published, this will include your full peer review and any attached files.

Reviewer #1: No

Reviewer #2: No

---

## [Decision Letter · Decision Letter 1]

2 Mar 2020

Dear Dr Grill,

We are pleased to inform you that your manuscript 'Kilohertz waveforms optimized to produce closed-state Na+ channel inactivation eliminate onset response in nerve conduction block' has been provisionally accepted for publication in PLOS Computational Biology.

Best regards,

Jonathan Rubin

Associate Editor

PLOS Computational Biology

Kim Blackwell

Deputy Editor

PLOS Computational Biology

Reviewer's Responses to Questions

**Comments to the Authors:**

Reviewer #1: My comments and suggestions have been fully addressed.

Reviewer #2: Great job addressing the comments!

**Have all data underlying the figures and results presented in the manuscript been provided?**

Reviewer #1: Yes

Reviewer #2: None

PLOS authors have the option to publish the peer review history of their article (what does this mean?). If published, this will include your full peer review and any attached files.

Reviewer #1: No

Reviewer #2: No

---

## [Editor Report · Acceptance letter]

8 Jun 2020

PCOMPBIOL-D-19-02003R1 

Kilohertz waveforms optimized to produce closed-state Na+ channel inactivation eliminate onset response in nerve conduction block

Dear Dr Grill,

I am pleased to inform you that your manuscript has been formally accepted for publication in PLOS Computational Biology. Your manuscript is now with our production department and you will be notified of the publication date in due course.

With kind regards,

Laura Mallard
